# Complexity of Sex Differences and Their Impact on Alzheimer’s Disease

**DOI:** 10.3390/biomedicines11051261

**Published:** 2023-04-24

**Authors:** Marion Kadlecova, Kristine Freude, Henriette Haukedal

**Affiliations:** Department of Veterinary and Animal Sciences, Faculty of Health and Medical Sciences, University of Copenhagen, 1870 C Frederiksberg, Denmark; marion@sund.ku.dk (M.K.); kkf@sund.ku.dk (K.F.)

**Keywords:** sex differences, microglia, sex hormones, estrogen, Alzheimer’s disease, hiPSC, cerebral organoids

## Abstract

Sex differences are present in brain morphology, sex hormones, aging processes and immune responses. These differences need to be considered for proper modelling of neurological diseases with clear sex differences. This is the case for Alzheimer’s disease (AD), a fatal neurodegenerative disorder with two-thirds of cases diagnosed in women. It is becoming clear that there is a complex interplay between the immune system, sex hormones and AD. Microglia are major players in the neuroinflammatory process occurring in AD and have been shown to be directly affected by sex hormones. However, many unanswered questions remain as the importance of including both sexes in research studies has only recently started receiving attention. In this review, we provide a summary of sex differences and their implications in AD, with a focus on microglia action. Furthermore, we discuss current available study models, including emerging complex microfluidic and 3D cellular models and their usefulness for studying hormonal effects in this disease.

## 1. Introduction

Alzheimer’s disease (AD) is a neurodegenerative disorder that is becoming a global health burden [1]. It is well documented that AD has a higher prevalence in women compared to men, as two-thirds of all cases are diagnosed in women. Consequently, women are, in general, at higher risk of developing AD, particularly post-menopause, suggesting an involvement of sex hormones in the disease [2]. However, clinical trials with hormone replacement therapy (HRT) have shown conflicting results [3,4]. This is not surprising though, due to the multifactorial origin of AD, and, therefore, sex hormones are most likely contributing factors rather than disease-causing factors. 

A significant caveat of most experimental studies as well as clinical trials is the persistent sex bias towards males within the medical field [5]. Although more women are diagnosed with AD than men, the experimental set up for research in AD often fails to take this sex difference into account and there is a general bias towards research studies and clinical trials conducted in males [2,6]. The same tendency is seen for research using animal models, particularly within the field of neuroscience, in which male mice have been shown to outnumber female mice 5 to 1 [7]. A large proportion of scientific articles have surprisingly not reported the sex of the animals used [8]. This bias is problematic given the known differences in cell population numbers, subtypes and localization between males and females. Such sex differences are particularly evident in microglia, but have also been demonstrated in neuronal cells [9], which could potentially contribute to the lack of sufficient treatment strategies at the moment. Recently, this bias has been increasingly recognized and taken into consideration in research designs. For instance, Nature Portfolio journals are now implementing new policies to promote sex and gender reporting in research studies [10]. Such initiatives will hopefully lead to better representation of both sexes in research in the upcoming years, contributing to novel insights into the role of sex in AD mechanisms to improve therapeutic interventions.

This review focuses on the main biological sex differences between men and women with regard to brain development and morphology, aging and immunological differences in the brain. We place a special focus on the complex involvement of sex hormones affecting the brain, their mechanism of action on microglia and what influence these have on AD pathology in women. Finally, we briefly summarize current available models for studying AD and discuss their applicability in studying sex differences and microglia involvement in this disease.

## 2. Alzheimer’s Disease

Dementia currently affects 50 million people worldwide with approximately 70% of cases caused by AD, making it the most prevalent neurodegenerative disorder [11]. AD is characterized by a progressive loss of neurons, particularly in the hippocampus and cortex, affecting memory, language and personality. To date, there is still a lack of treatments that can halt or reverse disease progression. Familial cases of AD (fAD) can be caused by specific mutations in either the amyloid precursor protein (*APP*), Presenilin 1 or Presenilin 2 (*PSEN1* and *PSEN2*). fAD only accounts for about 1–5% of all AD cases and usually affects individuals in their 40′s and 50′s and is therefore considered as early-onset AD [12]. Most AD cases are sporadic (sAD) with a late onset at the age of 60 and above and no apparent heritability. These cases are caused by a combination of various genetic single nucleotide polymorphisms (SNPs) and environmental factors [13]. Genome-wide association studies (GWAS) have identified genetic variants such as apolipoprotein E4 (*APOE4*), which, in its homozygous version, can increase the risk of developing sAD up to 15-fold [14]. More recent GWAS analyses have identified additional SNPs associated with an increased risk of developing AD (Table 1), though these represent a much lower risk compared to the APOE4 status [15,16].

The neuropathological hallmarks of AD—formation of extracellular amyloid beta (Aβ) plaques and intracellular neurofibrillary tangles (NFT)—are observed in both fAD and sAD cases [17,18]. Therapeutic strategies have traditionally been focused on targeting these Aβ and NFT pathologies, but promising candidates for intervention tend to fail in clinical trials. This could potentially be explained by the fact that Aβ and NFT phenotypes are characteristic for late stages of AD, and the pathogenesis is likely initiated decades before these hallmarks or any symptoms appear. This emphasizes the knowledge gap between initial disease mechanisms and late-stage pathology. Several potential early neuronal AD phenotypes have been described using multiple disease model platforms. These include hypometabolism, mitochondrial defects, synaptic dysfunction and hyperexcitability [19,20]. However, additional research into these early phenotypes, with consideration for brain-cell-specific pathologies, is needed to facilitate the development of compounds with early intervention potential. A recent paradigm shift has occurred in the field, acknowledging that non-neuronal cell populations, such as microglia and astrocytes, are significantly contributing to AD pathogenesis. Consequently, neuroinflammation is emerging as a key hallmark of neurodegenerative disorders, including AD, indicating an important role of the immune cells of the brain, particularly the microglia. Microglia are the resident macrophages of the central nervous system (CNS), which become abnormally activated in AD, highlighting their important role in disease pathology [21]. Microglial involvement is further supported by the fact that many of the identified risk SNPs for AD are located in immune-related genes, with restricted expression specific to microglia (Table 1). In light of this, recent efforts have been placed on studying the microglial contribution to AD [22,23].

## 3. Biological Sex Differences

### 3.1. Brain Morphology

An important factor to consider when studying neurological disease mechanisms are the differences in brain morphology observed between men and women and how this might play a role in a given disease. Taking these differences into account can help elucidate neurological disorders that affect men and women differently, such as AD. Already in utero, differences are present between male and female fetuses regarding functional connectivity of neural networks in cortical and subcortical areas of the developing brain [24]. At birth, males show an estimated 5.87% larger intracranial volume than females, and differences in grey matter volume (GMV) have been observed in various brain regions [25]. The overall raw GMV is higher in males, with a particular increase observed in the midbrain, left inferior temporal gyrus, right occipital lingual gyrus, right middle temporal gyrus and bilateral cerebellar hemispheres compared to women. In contrast, women show more GMV in areas such as the dorsal posterior cingulate, dorsal anterior cingulate, left cingulate gyrus and right inferior parietal lobule [26]. These structural brain differences were not directly correlated to androgen exposure or alterations in androgen receptor sensitivity. Longitudinal studies following sexual dimorphic traits of the brain during childhood, adolescence and early adulthood identified age-related differences in GMV in selected areas of the brain, with varied developmental trajectories resulting in GMV peaking on average two years earlier in females compared to males [27,28].

The difference in total brain volume increases to about 11% in adult men compared to women [29]. A meta-analysis identified sex differences seen especially in brain areas related to common neurological conditions such as the hippocampus, amygdala and insula. Males in general show larger brain volume across all subcortical regions, and these differences seem to be linked with the differences in total brain size [30]. While the connectivity in the brain varies from one individual to another, functional magnetic resonance imaging (fMRI) studies have shown general differences between men and women, with men showing greater connectivity in the visual and sensorimotor cortices and women showing greater connectivity in the default mode network (DMN) [31]. Interestingly, the DMN deactivation necessary for hippocampal activation during learning is impaired in AD patients, resulting in poor memory [32]. Moreover, patients with Autism spectrum disorders (ASD) or Attention deficit hyperactivity disorder (ADHD) show similar DMN abnormalities [33], which is interesting given the link between ADHD and the higher risk of developing AD [34]. The DMN has also been shown to be particularly vulnerable to Aβ plaque deposition [35]. Aβ accumulation, which is one of the earliest phenotypes of AD, has been suggested to follow a consistent spatiotemporal ordering beginning in cortical areas that overlap with the DMN. Aberrant DMN connectivity has been associated with amyloid burden and memory decline, which could thus contribute to the different risk profiles in women versus men [36]. Further sex-related differences in surface area, cortical thickness (CT), white matter microstructure and functional connectivity have been documented [29]. Interestingly, women present greater CT measures in many regions such as the bilateral cingulate cortex, bilateral temporal regions and left parietal regions, as well as more stable CT and memory performances compared to men. However, there is a rapid decline in this thickness from the mild cognitive impairment (MCI) stage to AD in women, particularly in brain areas most affected by AD. This was further correlated with Aβ status, indicating that sex differences in CT and memory can contribute to the different vulnerability of AD in women compared to men [37].

Several studies have been conducted to understand the underlying reasons for the differences observed between males and females in morphology and network connectivity of the brain. For instance, sex-biased regional GMV differences in the adult cortex have been correlated with differential regional expression of sex chromosome genes (compared to autosomal). The top-ranked X-linked and Y-linked genes correlating with these differences included *PCDH11Y*, *PCDH11X* and *PCDH19*, three genes from the protocadherin family that are involved in cell–cell recognition and development of the CNS, as well as *ZNF711*, a zinc finger protein transcription factor that is implicated in X-linked intellectual disability. Moreover, this transcriptomic analysis revealed other specific brain-expressed genes whose expression levels correlated with GMV sex differences. These included genes encoding transmembrane proteins such as *TMEMs*, *SLC44AC* and *TP53I11*, as well as genes important for cell growth, differentiation and CNS development such as *NELL1*, *SEMA5B* and *FRAT2*. These were all significantly upregulated in areas where GMV was greater in males and downregulated where GMV was greater in females [38]. Interestingly, this analysis did not show enrichment for gene sets related to sex steroid receptors or biosynthesis. Further, not only sex chromosome gene expression, as described above, but also sex chromosome dosage has been linked to regional heterogeneous changes in both grey and white matter [39]. Additionally, sex differences in brain morphology and connectivity are age-related [40]. In general, both sexes demonstrate a decline in GMV during normal aging, but males have been shown to have a greater volume loss over time compared to women in most brain areas, including the frontal, temporal and parietal regions. These findings suggest that women are less vulnerable to age-related atrophy, which is surprising given the high prevalence of AD in women compared to men. These findings indicate that, although anatomical differences in brain matter exist between males and females, this cannot be directly correlated to the increased risk of developing AD in females [41]. It has been postulated that the influence of sex hormones during brain development is a determinant in sexual dimorphism of brain morphology, and sex hormones could potentially play a role in brain atrophy [42]. Estrogen and progesterone have been suggested to have a protective effect against brain-volume loss in women, and the post-menopausal drop in these hormone levels could explain the increased vulnerability to neurodegeneration with age. Therefore, even though no enrichment of gene sets related to sex steroid receptors were identified in the previously discussed transcriptomic analysis, the possibility that androgens play an important role for the differences observed cannot be excluded, and this could render females more vulnerable to AD. This vulnerability could potentially be linked to the significant changes in hormone levels post-menopause, which, due to an overall reduced brain volume, might have a stronger impact in women.

### 3.2. Sex Hormones

Sex determination in humans is influenced by (1) genetic sex, i.e., the X and Y chromosome gene expression and (2) hormonal sex, which is dependent on sex hormone production for establishing phenotypic sex traits [43]. Dimorphic traits observed in anatomy, behavior and molecular function are all influenced by sex hormones. While both sexes share estrogens and androgens as sex hormones, the differential exposure to these leads to major biological differences, especially considering that hormonal influence can play a role not only during development but also acutely throughout life [44]. Male brains develop in utero under an initial surge of testosterone between gestational week 8–24 as a result of the activation of the testes, with a second surge of testosterone shortly after birth [45]. Following this, a significant rise in testosterone then only occurs at puberty, with the levels staying steady from there on out [44]. In contrast, women are not exposed to significant levels of estrogen until several days after birth, with an estrogen rise linked to activation of steroidogenesis in the ovaries [46]. Furthermore, women go through two major transitional sex hormone cycles at puberty, with a cyclic pattern of exposure to estrogens during menstruation and at menopause, when there is a significant drop in estrogen levels (Figure 1) [47].

While the main source of 17β-estradiol (E2), the most potent and prominent form of estrogen, comes from the ovaries, both neurons and astrocytes can produce E2 locally in the brain from androgen precursors via aromatase enzyme action [48]. Under homeostatic conditions, the activity of aromatase is demonstrated mainly in neurons, with a number of factors including sex, age, hormones and neurotransmitters implicated in the regulation of brain aromatase levels [49]. However, in conditions of brain injury or inflammation, aromatase activity is induced in astrocytes, indicating increased estrogen production [50,51]. Estrogen plays various regulatory roles in important processes such as synaptic plasticity, neuronal growth, memory formation and neuroprotection [49]. The precise mechanisms behind the neuroprotective effect of estrogen are not completely understood, but it seems to mediate an increase in growth factor production and synapse formation as well as antioxidant and anti-inflammatory pathway activation [52].

Steroid hormone receptors, found in neurons throughout the whole brain, can either be classical nuclear receptors, acting as transcription factors directly influencing gene expression, or non-nuclear membrane-associated receptors, acting through faster, non-genomic mechanisms via protein kinase cascades [53]. Steroid receptors have now also been identified in glial cells such as microglia and astrocytes, indicating that non-neuronal cell populations are also direct targets of such hormones [54]. The main E2 receptors are nuclear estrogen receptors (ER)α and ERβ, while G-protein coupled estrogen receptors (GPER) such as GPER1 are common membrane-bound receptors. ER expression varies with age, sex and amounts of circulating hormones. Testosterone and dihydrotestosterone are the main ligands for nuclear androgen receptors (ARs) [55]. Furthermore, testosterone can be directly converted to E2 via aromatase, which further binds ERs. Aromatase expression has been shown to be higher in males in all brain regions, and this is thought to compensate for the lower levels of circulating estrogen [56]. Interestingly, polymorphism in the *CYP19A1* gene encoding aromatase has been associated with increased AD risk, hinting to an important role of E2 regulation in AD pathology [57].

Both estrogen and testosterone play a role in learning and memory formation, mediated through alterations of dendritic spine density [58]. Dendritic spine plasticity underlies memory formation, and increased density has been associated with long-term potentiation, particularly important for learning and memory [59]. Estrogen contributes to increased dendritic density through interactions with the extracellular signal regulated kinase-mitogen activated protein (ERK/MAP) kinase pathway. Testosterone can act through the same mechanism after being converted to E2 [60]. Alternatively, testosterone can act via the ARs, which activate cortical pathways involved in spatial cognition and mood [61].

Many rodent studies have focused on the role of E2, specifically in the hippocampus, where there is an abundance of ER subtypes in both males and females. Here, E2 largely mediates hippocampal function in both sexes, but through different molecular pathways. It significantly enhances hippocampal synaptic plasticity in both females and males. However, in females, this is mediated by a pre-synaptic increase in glutamine release, regulated by Erβ, and a post-synaptic increase in glutamate sensitivity, mediated by GPER. In contrast, in males, ERα regulates pre-synaptic glutamate, and ERβ mediates post-synaptic sensitivity [62]. Interestingly, the abundance of both ERα and ERβ decreases during aging, indicating that the neuroprotective effect of estrogen is reduced. Moreover, since ERs mainly bind to estrogen, a reduction in this hormone might lead to additional risk, explaining why the menopausal drop in estrogen levels leads to increased vulnerability to pathological events such as AD [63].

Several studies have indicated that HRT can lower the risk of developing AD in menopausal and post-menopausal women, suggesting a protective role of estrogen in AD in women [64,65]. However, other studies could not confirm a causal relationship between HRT and AD risk [4,66]. As with many intervention strategies, the genetic and environmental risk factors need to be carefully included in a personalized medicine strategy to determine the beneficial effects of HRT. For instance, the beneficial effect in selected patient groups has been shown in a recent study wherein the authors investigated the impact of age on HRT initiation according to APOE4 carrier status [67]. Similarly controversial is the depletion of estrogen via tamoxifen treatment in breast cancer patients. Whilst some studies suggest a link between estrogen depletion and risk of AD [68], others cannot support such a link [67] or even find that tamoxifen can protect against AD [69]. To summarize, estrogen has been proven to be neuroprotective, but the effect on estrogen depletion and AD development remains controversial. Recent studies aiming at better stratification of patient cohorts, including APOE4 carrier status, are promising to determine the patient subgroups, which could benefit from HRT.

### 3.3. Aging

Aging is characterized by a progressive decrease in normal physiological function accompanied by a wide range of changes in genetic expression and stability, intercellular signaling, immune system response and cognitive ability [70,71]. Aging is further associated with chronic inflammation, with inflammatory modulators most likely possessing a cross-play with other aging mechanisms such as cellular senescence [70]. Brain metabolism gradually declines, shifting to a predominantly oxidative glucose metabolism. The relative aging trajectory of the brain varies between individuals, but, in general, females show a younger metabolic brain age at every stage of adulthood compared to men [72]. However, it seems that they experience earlier aging-related gene expression changes with faster manifestations compared to men, resulting in lower energy production and neural function as well as increased immune activation [73]. Despite showing a relatively younger metabolic brain age, aging mechanisms could potentially have a stronger impact on women and therefore also increase their vulnerability to AD.

Cognitive decline and brain aging is associated with oxidative damage as neurons accumulate damaged mitochondria and proteins. Furthermore, neurons suffer from energy and nutrient imbalances, disturbing calcium homeostasis, which is crucial for normal neuronal function. Hippocampal neurons are particularly vulnerable to calcium excitotoxicity, which explains cognitive decline as one of the features of aging. With decreased energy production, neurons are more likely to suffer from synapse alteration and degeneration [74].

Normal brain aging mechanisms coincide with a decrease in sex hormones, so it is not surprising that these two have been linked. Hormonal changes associated with aging in men are debated, with some studies stating that total testosterone levels stay relatively stable [75], whilst other studies indicate that there is a gradual decline in testosterone in aging men [76]. In contrast, the rapid loss of estrogen in women is well documented. Given the important roles of sex hormones in the brain discussed in the previous section, it is not surprising that the difference in gradual compared to rapid loss of sex hormones in men and women, respectively, will have divergent effects on brain aging [77]. ER expression has been shown to vary with aging [55], though with some contradictory findings. A study from Japanese monkey hippocampi showed that ERβ expression increases as a result of natural menopause, while surgical menopause results in an increase in aromatase expression with no changes in ERβ [78]. Another study using female rats showed that ERβ mRNA levels decreased with age in a brain-region-specific manner, which was, however, difficult to study since the hormonal environment was not controlled due to high variance. On the other hand, ERα mRNA levels seemed to be relatively unaffected by aging [79]. Discrepancies in results are most likely due to the different study models used as differences in ER subtype expressions in different brain regions have been shown to exist between species. For instance, humans and primates predominantly express ERβ in the hippocampus [80,81], while rodents express ERα [82,83]. Given that these two ER subtypes have been shown to affect gene expression and neuronal function differently, more studies are needed to fully understand changes in expression during aging, especially in males, which has not been sufficiently studied thus far.

The impact of aging and sex on the immune system is still relatively unexplored, but differences between men and women in varying numbers of immune cell subpopulations, cellular communication and gene expression have been identified (Figure 1) [84]. Increased phagocytic activity has been identified in aged female microglia compared to males [85]. Furthermore, female microglia lose their ability to adapt to an inflammatory environment faster than male microglia. This can potentially be a contributing factor to the increased vulnerability of women to late onset neurological disorders involving the immune system. An increase in hippocampal and entorhinal cortex immune and inflammatory gene expression is observed with aging in both sexes, but with women also displaying increases in other brain regions compared to men [86]. Females generally seem to show a more pro-inflammatory brain phenotype with aging, which is closely associated with menopausal drops in estrogen levels [87], likely linked to the loss of the modulating effect of estrogen, which normally promotes the anti-inflammatory action of microglia [88].

### 3.4. Sex Differences in Alzheimer’s Disease

In AD, there is a significant decrease in cortical thickness and surface area in brain regions associated with cognition [89]. Both age and sex are major risk factors for the development of the disease [90]. Compared to normal age-associated decline in GMV, patients with AD show a significant reduction in grey matter in the hippocampus and entorhinal cortex [71]. Identifying the precise sex-related factors that contribute to the increased risk of AD in women compared to men has proved to be challenging. However, the drop in estrogen following menopause has been implicated as a major player [91], supported by the fact that post-menopausal women represent about 60% of AD patients. This has been termed the “estrogen hypothesis”, which postulates that the risk for AD is correlated with the loss of the neuroprotective effect of estradiol [90]. The deterioration of cognitive abilities has been shown to be worse in female AD patients [92]. Furthermore, brain imaging analysis identified the emergence of AD biomarkers—glucose hypometabolism, Aβ depositions and brain atrophy—to be positively correlated with the peri-menopausal and post-menopausal stages in cognitively normal women compared to men of the same age [93]. Additionally, women might be more vulnerable to genetic risk contribution as the APOE4 allele has been shown to confer great risk in female carriers compared to male carriers [94]. Furthermore, female AD patients are usually diagnosed at advanced ages and, therefore aging mechanisms, as previously discussed, most likely play an additional role in the increased vulnerability observed in women.

It is important to note that study design, sample selection criteria and many other factors can greatly influence the outcome of experiments, which might explain some of the inconsistent findings and difficulty understanding certain aspects of AD, given how many factors can affect the risk and progression of the disease [95]. For instance, in addition to the hormonal and genetic contributions to sex differences in AD, many factors influence the disease, e.g., pregnancy, co-morbidities and neuroinflammation [95,96]. Neuroinflammation has gained more attention with the emergence of microglia as key players in the disease [97]. Further exploring the combined role of age and sex on the immune system could help our understanding of the different susceptibilities of men and women to AD pathology.

## 4. Microglia

Microglia are the dominant resident macrophages of the brain and are responsible for a wide range of immune and homeostatic functions throughout life. Microglia develop in the yolk sacs of mesodermal hematopoietic cells [98,99] and migrate to the brain around embryonic day 9 in mice [100] and between gestational weeks 2 and 8 in humans until the formation of the blood-brain barrier [101]. They help regulate neurogenesis as they localize more densely in areas of active cortical formation, phagocytosing neural precursor cells. Any alterations to microglial activity or quantity can therefore have detrimental developmental and behavioral consequences [102]. During early postnatal development, microglia contribute to shaping neural networks by participating in synaptic pruning. Signaling via the fractalkine receptor Cx3cr1 attracts microglia and mediates elimination of dendritic spines and excess excitatory synapses [103]. The complement cascade components C3 and C1q have also been shown to mediate microglia engulfment of excitatory inputs [104,105].

Microglia are a heterogeneous cell population with highly plastic phenotypes, and their function is determined by their locations in the different regions of the brain and local stimuli from the surrounding microenvironment [106]. Under normal physiological conditions, microglia possess small cell bodies with highly ramified and motile processes that surveil the microenvironment of the brain for pathogens and injuries while cleaning up debris and waste products [107]. Under pathological conditions, they shift morphology to general amoeboid cell forms with retracted processes and large cell bodies. However, they also show a variety of transitional states that are most likely disease-determined [108].

In healthy adult brains, microglia dynamically interact with the other cell types in the CNS. They rely on signaling from neurons, astrocytes and oligodendrocytes for their maturation, identity and survival [109]. The importance of these signals was demonstrated in ex vivo experiments of microglia, which showed complete alteration of their characteristic gene expression [110,111]. Reciprocally, microglia interact with neurons to maintain synaptic function and modulate synaptic plasticity. Microglial secretion of brain-derived neurotrophic factor (BDNF) is a key molecule involved in synaptic plasticity. Depleting BDNF signals resulted in altered synapse formation and deficits in learning and memory in mice, showing a similar effect to a complete microglia depletion in the brain [112].

While microglia provide support and protection for the CNS, chronic activation can become harmful as their production of reactive oxygen species, inflammatory cytokines, complement proteins and proteases damage neighboring cells [113]. In aging, microglia adopt a more pro-inflammatory state accompanied by changes in their morphology, impairments in motility, processing speed and loss of neuroprotective effects [114]. These age-related changes therefore have important consequences for the microglial ability to cope with late-onset diseases, and they are implicated in neurodegenerative disorders. Under pathological conditions, microglia are the main drivers of neuroinflammation, with complex mechanisms of action resulting in both beneficial and adverse effects, determined by the type and duration of the insult [115]. Adding to the complexity of microglial actions, their responses in aging and disease are highly influenced by sex [116].

### 4.1. Male and Female Microglia

One of the obvious differences between male and female microglia, as with all other cell types of the body, is the expression of the sex chromosomes. The X chromosome is known to carry the largest amount of immune-related genes [117] and while X inactivation occurring in females should result in equal expression of these genes in both sexes, some genes such as Toll-like receptor 7 (*TLR7*) involved in type I interferon signaling escape the inactivation, resulting in biallelic expression of certain immune genes in females [118]. Multiple other genes have been suggested to escape X chromosome inactivation such as *KDM5C*, *KDM6A*, *PCDH11X*, *CYBB*, *RPS6KA3*, *IR13RA1*, *BTK*, *CXCR3*, *USP11*, *CD99* and *USP27X* [119,120,121]. These are all involved in immune-related pathways, and X chromosome gene dosage could therefore contribute to intrinsic differences in female microglial responses [122].

Microglia play an important role in sex-dependent differentiation of the brain during development. This is correlated to prenatal hormone exposure, which changes the general microenvironment and drives sex-specific phenotypes in microglia [123]. For instance, masculinization of the brain is entirely dependent on microglia mediating estradiol-induced upregulation of the inflammatory mediator prostaglandin E(2) and results in increased density of dendritic spines and male-specific behavior [124]. Transplanted microglia from one sex to the other resumes their normal function, but retains their sex-specific genetic signature, showing how important development and early exposure to sex hormones are for sex-specific differences in male and female microglia [125]. Moreover, transcriptomic analysis in mice found that microglia follow different developmental trajectories, with female microglia transcriptomes more developmentally advanced at P60 compared to male microglia [126]. Intriguingly, microglial transcriptomic analysis in brain tissue from AD and autism patients identified significantly accelerated microglial development compared to controls, pointing towards the important involvement of microglial developmental trajectories in neurological diseases.

Many studies have found sex-specific differences in the quantities and morphologies of microglia in various brain regions [123,127] as well as in their transcriptomes [125]. Under physiological conditions, male microglia are generally more reactive and possess larger soma and increased motility, while female microglia are more phagocytic and show higher expressions of genes related to cellular repair and inflammatory regulation [128]. Regarding aging, female microglia show higher expressions of inflammation-related genes compared to males, as observed in mice hippocampi and cerebral cortices (Figure 2) [129]. Furthermore, microglial responses to injury and disease differ between males and females, with sex hormones playing an important role in reducing microglial inflammatory responses. For instance, cell death because of injury, such as stroke, is mediated by caspases in females, whereas the process is commonly dependent on Poly (ADP-ribose) polymerase 1 (PARP-1) and the nitric oxide (NO) pathway in males. Moreover, females demonstrate higher levels of microglia with an anti-inflammatory phenotype post-injury, which is associated with reduced inflammation. In contrast, male microglia show increased TLR2 signaling, indicating increased inflammation [130]. These differences clearly indicate the important role of microglia and innate immunity in the sexual dimorphism that exists in brain trauma and neuroinflammatory conditions.

### 4.2. Microglia and Estrogen

As stated above, sex hormones play an important regulatory role in microglia and mediate their sex-specific phenotype. Particularly, estradiol is an important microglial modulator, acting on all three main ERs: ERα, ERβ and GPER1 [131]. Estrogen binding to either ERβ or GPER1 in microglia has been shown to have strong neuroprotective effects (Figure 3) [132]. This is mediated by the annexin A1 (ANXA1) protein, a potent anti-inflammatory mediator [133], and results in increased microglial phagocytosis of apoptotic cells and microglia adopting an anti-inflammatory phenotype as a response [132]. The anti-inflammatory action of estrogen was further shown in multiple studies with LPS-treated microglia, where E2 was able to block the production of inflammatory mediators [125,134,135,136]. Furthermore, sex-related differences in E2-action microglia have been shown. For instance, LPS treatment of isolated neonatal rat microglia induces more mRNA expression of pro-inflammatory interleukin 1β (IL-1β) in males than in females. Treating these with E2 resulted in anti-inflammatory effects in males, whilst pro-inflammatory responses were observed in females. The opposite was seen in microglia isolated from an adult rat’s hippocampus, where E2 dampened the LPS-induced IL-1β response in females but not in males [137].

A study using young and aged ovariectomized mice showed that the combined effect of aging factors and estrogen depletion exacerbates the increased inflammation normally associated with aging, as this was not seen in the young ovariectomized mice [138]. This was observed together with age-dependent transcriptional upregulation of inflammatory mediators and microglia morphological changes. Another study using ovariectomized rats observed a significant increase in phagocytic and toll-like receptors including Cd11b, Cd18, C3, Cd32, Msr2 and TLR4 [87], once again proposing an increased reactivity of microglia because of estrogen depletion. While surgical ovariectomy cannot be compared to the normal process of estrogen decrease during human menopause, surprisingly similar results were observed in humans [86,87], showing the significant role of estrogen in modulating the neuroinflammatory mechanisms of microglia.

### 4.3. Microglia, Neuroinflammation and Alzheimer’s Disease

Since Aβ plaques and NFTs alone do not explain the development of AD, many studies have now established neuroinflammation as another important driver of AD pathology along with evidence of a sustained inflammatory response in post-mortem brain tissue from AD patients as well as in pre-clinical models [139]. In vivo imaging studies from AD patients and mice have correlated microglia activation to cognitive decline and have shown that this activation occurs relatively early and varies during disease progression [140]. Furthermore, blocking microglial reactivity was shown to alleviate synaptic deficits and cognitive impairment in AD mice during later stages of the disease [141].

In early stages of the disease, activation of the immune system can play a protective role, as microglia migrate to regions of Aβ deposits, where they work to degrade and remove these [142], a mechanism shown to be driven by TREM2 activation [143]. However, Aβ acts on the TLR2 microglial receptor, inducing the production of pro-inflammatory molecules, indicating that, in response to prolonged exposure to Aβ plaques, microglia become chronically activated and toxic to their surrounding environment [144]. This dysfunctional phenotype of microglia observed in late AD has been associated with the activation of TREM2, following phagocytosis of apoptotic neurons, which triggers APOE signaling and results in the suppression of the homeostatic functions of microglia [145]. Furthermore, microglia contribute to the spread of tau protein from the entorhinal cortex to the hippocampus via exosome secretion [146]. This is likely also mediated by TREM2, as knocking out TREM2 in mice resulted in an amplified spread of tau and correlated with synaptic loss and cognitive impairment [147]. Furthermore, pro-inflammatory microglia have now also been shown to contribute to tau phosphorylation, exacerbating the general tau and Aβ pathology [148]. Tau oligomers and monomers, actively released by neurons, can act on microglia and activate the NLRP3 inflammasome, which in turn increases IL-1β secretion, regulating tau kinases and therefore driving tau hyperphosphorylation [149].

The sexual dimorphism observed in female prevalence of AD can potentially be related to sex differences in microglia. For instance, increased amyloid load was observed in female post-mortem brain tissue compared to decreased load observed in male patients. This correlates with observations from AD-mice models, where female microglia were less phagocytic and more reactive then male microglia [150]. Furthermore, ramified microglia confer neuroprotection in the hippocampus under excitotoxic conditions in pathology [151], and it has been observed that there is a significantly lower amount of ramified microglia in women compared to men in response to Aβ42 [152].

Transcriptomic analysis of microglia nuclei isolated from post-mortem tissue revealed an enrichment of AD risk genes and genes related to the inflammatory responses in healthy post-menopausal and AD females [153]. Ovariectomy in female AD mice accelerated the activation of microglia that surrounded Aβ plaques, which was reversed with E2 administration [154]. It is thus suggested that E2 can increase Aβ protein uptake by microglia, providing early protection in AD [155]. E2 also reduced the number of Aβ plaque-bearing microglia, as observed in a mouse model of AD [154].

Taken together, these findings strongly suggest a bias in risk for women to develop AD, with microglia playing an important role. Many aspects of AD pathology as well as the precise role of microglia remain to be fully investigated in order to gain a better understanding of the underlying disease mechanisms as well as the implication of sex. However, this requires adequate disease models capable of mimicking the complex cellular and molecular interplays occurring during this disease. Many of the current observations and findings were obtained in rodent models, which differ in their immune systems in terms of both innate and adaptive immunity. Such discrepancies include expression of TLRs, cytokines, chemokines and cytokine receptors as well as differences in various signaling pathways [156]. Although the global transcription profiles are conserved between the two species, several gene sets clearly show divergent expression in mice compared to humans. This is particularly evident in immune cell populations, observed by differential expression of genes such as *CD38*, *CD56*, *TMEMs* and *IL15* [157]. The discrepancies in murine versus human immune systems makes it challenging to study Aβ plaque load and tau seeding consequences and should be considered when using rodents as models for human diseases. Moreover, most research has been conducted in male rodents, neglecting sex differences in AD, further highlighting the need for new studies and additional model systems.

## 5. Common Model Systems for Studying Alzheimer’s Disease

### 5.1. Mouse Models

Murine Aβ peptides do not naturally form plaques due to a difference in three amino acids in the coding sequence compared to humans [158]. Attempts to construct knock-in mouse models, where these three amino acids were replaced by the human equivalent, did not result in significant AD phenotypes [159], which is why human transgenes are usually used to express the phenotype of Aβ plaque accumulation in mice [160,161]. The most commonly used mouse models are the ones carrying human mutations from fAD in *APP* and *PSEN1* as the research focus has been on the two main pathological mechanisms of Aβ plaques and NFTs in AD [162]. Some of these mice, such as the Tg2576 mouse line overexpressing the K670M/N671L Swedish mutation [163], carry a single human mutation, while others, such as the 5xFAD mouse line expressing the *APP* Swedish K670M/N671L, Florida I716V, London V717I as well as *PSEN1* M146L and L286V mutations [164], carry multiple human mutations. What all of these different mouse models have in common is the progressive development of Aβ plaques and memory deficits; however, the most significant drawback is the lack of NFTs and neurodegeneration. To model tau pathology, mice carrying the human microtubule associated protein tau (*MAPT)* transgene have been generated, but these present with a frontotemporal dementia-like pathology rather than AD [165]. These models have therefore been crossed with AD-mice models to obtain the combined pathology of plaques and NFT [166].

Importantly, none of the models generated thus far recapitulate all the pathological hallmarks of AD. Each model shows only some aspects of the pathology and to varying degrees [167]. Furthermore, these models only allow the study of the familial form of AD and not the more prevalent sporadic forms, which have no clear genetic link [168]. Unfortunately, most clinical trials based on successful studies conducted in AD-mice models have failed or shown only temporary symptomatic relief for AD patients [169,170]. There are many factors that can affect the translation from pre-clinical to clinical trials, some of which include inter-species differences in disease mechanisms or disease progression, drug metabolism and anatomy.

### 5.2. In Vitro Models

#### 5.2.1. 2D Cellular Models

A variety of in vitro cellular models have been generated for the study of AD [171], most often using neurons derived from human induced pluripotent stem cells (hiPSCs), reprogrammed from somatic cells of patients with fAD and sAD or using CRISPR/Cas9 technology to knock-in genes of interest [172,173,174]. These neurons have shown increased production of Aβ and tau hyperphosphorylation [175,176,177,178]. However, like mouse models, these hiPSCs-derived models do not recapitulate all pathological hallmarks of AD. For example, the increased Aβ production is not enough for plaque formation. Since cultured neurons lack the maturity of in vivo neurons due to limited culturing time, it is understandable that this age-related aspect of the disease is challenging to recapitulate [179]. In contrast, the monoculture of specific neuronal subtypes with little to no glia support can trigger the expression of disease phenotypes, such as increased phosphorylation of TAU protein, accompanied by increased expression of active glycogen synthase kinase 3 β, as well as significant changes in Aβ 40/42 [180]. These results have led to investigation of disease pathology in other cell types relevant for AD, generated from hiPSCs, such as microglia and astrocytes [171]. The former has been shown to phagocyte AD-related products, such as Aβ and tau oligomers, and significantly upregulate AD risk genes once exposed to Aβ [181].

Even though disease-specific phenotypes can be documented and investigated in a cell-type-specific manner, it remains to be seen if these phenotypes persist or are altered in more complex co-cultures. Some of the obvious limitations of using monolayers compared to the human brain include the lack of a complex microenvironment, interactions with other neuronal and non-neuronal cell types and extrinsic signals. The complexity can be increased to a certain extent with co-cultures or tri-cultures of neurons, astrocytes and microglia [182,183], allowing the study of cell-to-cell interactions. Astrocytes showed greater physiological morphology, microglia were less inflammatory and neurons developed more branching and post-synaptic markers. This kind of set-up also shows increased potential to study microglia activation, which relies on astrocytes and neurons. Furthermore, it could be used as a screening platform for therapeutics specifically targeting cell interactions. Limitations remain as this model system lacks vascularization and barriers, such as the blood-brain barrier, which would limit the access of extrinsic factors to the cells under in vivo conditions.

#### 5.2.2. Cerebral Organoids

The limitations of mouse models and cell monolayers have led to the development of cerebral organoids as a study model for neurodevelopment and neurodegenerative diseases (Figure 4). Cerebral organoids mimic human fetal brain development via self-assembly and differentiation of stem cells to discrete brain-like regions, showing cortical layer structures with mature neurons, neuronal precursors and astrocytes [184]. These can form either in an unguided manner, relying on intrinsic cellular signals for differentiation, or in a guided manner, with growth factors and inhibitors directing the formation of specific brain regions, resulting in structural specificity, more reproducibility and specific cellular composition [185]. This model avoids the problem of inter-species differences and offers great potential for personalized medicine.

Several studies have now used cerebral organoids as study models for AD, using hiPSCs carrying mutations for both fAD and sAD [179]. Collectively, these studies have reported increases in Aβ production and formation of aggregates, increased tau phosphorylation and NFTs, endosomal abnormalities, synaptic loss and hyperexcitability [186,187,188,189,190]. Additionally, treatment with β- and γ-secretase inhibitors successfully reduced hyperphosphorylated tau and Aβ aggregates in these organoids. A large-scale generation of these AD-derived organoids has also been used as a drug screening platform for approved AD drugs [191].

Some of the limitations of this model include the lack of vascularization, appearance of necrotic core, low reproducibility and lack of oligodendrocytes and immune cells [185]. However, significant progress has been made with the incorporation of vasculature into organoids, either by transplantation into mouse brains [192] or via co-cultures with mesodermal progenitors [193]. Furthermore, the lack of immune cells has also been remedied with protocols establishing either co-cultures of organoids and microglia [194,195] or intrinsically developing microglia in the organoid during the differentiation process [196]. This leads to better maturation of the neurons, adds complexity to the model system and offers the opportunity to study immune-related mechanisms.

## 6. Discussion

In the past few decades, many different cellular and animal models have been established to study the complex pathological mechanisms involved in AD. Some individual aspects of the disease—an increase in Aβ, tau phosphorylation and cognitive impairment—have been recapitulated in these models and have been the focus of research. Compared to the intricacy of the human body, these “simplified” models are not comparable, and it is therefore not surprising that disease modelling is limited. New research avenues have emerged with the discovery of microglia and their implication in AD pathology. Moreover, it is now a generally established consensus that men and women possess divergent risks and phenotypes related to AD, with women being disproportionally affected. Female (XX) microglia display enhanced phagocytic activity and altered cytokine and chemokine secretion compared to male (XY) microglia. These differences result in divergent responses in AD in terms of ability to clear and respond to Aβ oligomers and plaques [116]. Recent pharmacological intervention strategies have targeted microglial function and disease-related responses through immunomodulary treatments [197]. Consequently, sex differences in microglial behavior and responses need to be fully explored to not only secure drug efficacy, but also to determine potential sex-specific side effects of immunomodulary treatments. However, the question remains how to study microglial functions and implications of sex and sex hormones in AD with the currently available models.

Due to limited possibility for human studies, animal models have been extensively used in the hopes of understanding pathological mechanisms of AD in an in vivo setting. Much of the knowledge we have about microglial origin, development and mechanisms of action stems from mouse studies. Certain limitations of mice models need to be considered when interpreting the findings of these studies. Important considerations include the size and architecture of the mouse brain compared to the human brain. Single nucleus RNA sequencing has shown significant divergence because of cell composition, laminar distribution and morphological differences [198]. Although more than 50% of homeostatic microglia genes are conserved between humans and mice, there are human-specific microglial genes not observed in murine models, such as *APOC1, MPZL1, SORL1, CD58, ERAP2, GNLY* and *S100A12,* which are genes related to the innate immune system [199,200]. Difficulty in comparing microglial data from AD patients to AD-mice models can also be the result of the inaccurate disease phenotype in mice. Furthermore, overexpressing human mutations in mice can be problematic. For instance, in an analysis of hippocampal transcriptome in 5xFAD mice, female mice showed more severe transcriptional upregulation of immune associated genes. However, these female mice also showed elevated expression of the 5xFAD transgenes, which could have accounted for the sex-related differential upregulation in transcriptome [201].

Another drawback of mice models is that the estrous cycle in rodents, the human equivalent of the menstrual cycle, which lasts for 4–5 days compared to 28 days for humans [202], is less defined, and the estrogen and progesterone peaks considerably overlap [203]. Aging female mice do not experience the equivalent of human menopause. They experience irregular cycles more representative of perimenopause [204]. Mice are therefore usually ovariectomized for the study of post-menopausal effects, but this will not mimic the gradual physiological decline of sex hormones in women, which have an impact on AD risk and microglial networks. Furthermore, the majority of AD studies using mice models have been conducted using male mice and have thus not accounted for the sex bias observed in AD. However, recent efforts have been made to meet the need of animal models that enable investigation of the effect of sex differences in AD. This includes the Four Core Genotype and XY* models, which can be useful to determine whether the causes of sex differences are hormonal or related to sex chromosome gene expression and dosage [205].

The generation of hiPSC and the establishment of differentiation protocols for various cellular subtypes of the CNS has greatly advanced our understanding of the human brain. This method offers great advantages for studies at a single cell level in a highly controllable environment. However, in the study of complex pathological mechanisms, it is impossible to study cell-to-cell interactions, molecular signals and effects of the microenvironment. Moreover, the outcomes of reprogramming somatic cells into a pluripotent state are highly variable. There are mixed data regarding whether hiPSCs keep the epigenetic signature from their donor, and it is unclear how this may affect their differentiation potentials [206]. Different clones from the same generated hiPSC line can possess differences in these epigenetic marks, thereby affecting their individual differentiation potentials [207]. Additionally, the reprogramming process further resets many aspects of natural aging, making it difficult to study age-related disorders [208].

There is one important consideration for using stem cells from either female or male donors. While both are reprogrammed to the pluripotent state, one important difference will remain—the presence of X and Y chromosomes. For studying sex-chromosome-related differences, hiPSCs offer the advantage of relatively easy genetic manipulation and knockout experiments [209]. However, in both embryonic (hESCs) and hiPSCs, there is heterogeneity in the process of X-chromosome inactivation in female-derived cell lines [210]. Those that do undergo X-inactivation can experience a phenomenon of “erosion” in culture over time, which results in transcriptional reactivation of silenced genes on the inactive X-chromosome [211]. With the X-chromosome carrying many immune-related genes, this can have significant implications for studying sex-related differences in AD.

Comparing male and X-inactivated female hESCs and hiPSCs revealed that reprogramming induced the expression of the SRY gene, present on the Y chromosome, which in turn could affect autosomal gene expression and differentiation potential in the male-derived stem cell lines. Furthermore, 227 sex-specific autosomal genes were found to be differentially expressed between female and male hESCs. There was a significant enrichment for genes associated with steroid metabolism, and estrogen was proposed to play an important role in the female pluripotent stem cell state and in further neuronal development. This was confirmed by the observation that estrone administration significantly increased the proliferation of female ESCs while leaving the male ESCs unaffected [212]. This stresses the importance of using both male- and female-derived cell lines in experimental studies to ensure that sex-specific differences related to sex chromosomes are not disregarded.

Many studies have agreed that primary microglial cultures are poorly representative of microglial action in vivo. Isolated tissue microglia need to be analyzed quickly as in vitro culture or even cryogenic storage significantly changes their phenotype, prohibiting long-term studies [111,213]. Microglial response to oligomeric or fibrillary Aβ stimulation in vitro also results in a very rapid, significantly greater response and transcriptional changes not observed in in vivo mice [214]. Several protocols for microglial differentiation from hiPSC have been established in the past few years as alternatives to primary cultures [215]. While the study of age-related mechanisms remains challenging, as hiPSC-derived microglia seem to have a fetal rather than adult phenotype, it offers the opportunity to use them in more complex co-cultures. Indeed, microglia co-cultured with cortical neurons already show more physiological phenotypes [216]. Even a tri-culture system of hiPSC-derived microglia, astrocytes and neurons carrying the *APP* Swedish mutation have now been used to study neuroinflammation in AD [183]. This shows the importance of studying cell-to-cell interactions as microglial crosstalk with astrocytes leads to increased complement C3 production, which plays a major role in synaptic loss during neuroinflammation.

A major step towards complex disease modelling has been the emergence of cerebral organoids capable of recapitulating human neurodevelopment and different regions of the cerebral cortex. Certain aspects of cerebral organoids remain to be further investigated, such as the fact that they possess the equivalent of a fetal transcriptome [217] and how this will apply to disease modelling, for example, in the case of late-onset AD commonly seen for sporadic forms [179]. Further difficulties accompany batch-to-batch variation and low reproducibility [218] and relative lack of vascularization, limiting the time these can be kept in culture [219]. Many of these issues are, however, being actively addressed, with optimized protocols and new study platforms.

Cerebral organoids show great potential in modelling sex-related differences if the right extrinsic signals are provided during differentiation. For instance, no significant differences were observed between cerebral organoids generated from both male and female ESCs, suggesting that sex chromosomes alone are not driving the developmental differences observed in the human brain. However, after treatment with sex steroids, there was a significant increase in cortical neural progenitor cell numbers in the male-derived cerebral organoids, which resulted in a thicker upper layer of excitatory neurons, and therefore morphological differences in male- compared to female-derived organoids [220]. This is most likely more representative of the differential developmental pathways occurring in female and male human brain development, during which the importance of sex hormones has been established.

Furthermore, since microglia play a crucial role in brain development and sexual differentiation [113], incorporation of these into cerebral organoids as a general practice will most likely lead to models that are more representative of the human brain. Microglia in cerebral organoids were shown to cluster around neural stem cells and induce transcriptional changes, as would be expected with their role in neurogenesis. Furthermore, they accelerate neuronal maturation and network activity. Microglial phenotypes within the 3D structure of the cerebral organoid mimic to a larger extent the in vivo microglial phenotypes compared to all other models [221].

Microfluidic platforms are also emerging as powerful tools for more-complex tissue cultures, with highly controllable microenvironments, continuous nutrient supply as well as waste removal [222]. While this has not yet been applied to cerebral organoids specifically, a microfluidic culture system has successfully been used to provide vascularization to organoids [223]. Furthermore, a microfluidic system has been established to model the whole female reproductive tract and menstrual cycle [224], showing great potential to be used in the future to study the role of sex hormones in normal physiological as well as pathological conditions.

Combining all available methods to establish a cerebral organoid model containing microglia in a microfluidic device with supporting vasculature and timely exposure to physiological levels of sex hormones would represent tremendous potential for the study of sex differences and microglial implication in AD (Figure 5).

To investigate female estrogen levels’ implication on development, female-derived cerebral organoids would have to be cultured at least 250–300 days, at which point they reach a “post-natal” stage [225], when estrogen could be administered to mimic female post-natal estrogen exposure. This could then be compared to male-derived cerebral organoids, exposed to testosterone during the “embryonic” stage, as described previously [220], together with a later surge corresponding to post-natal testosterone exposure. Such long-term experiments would be greatly facilitated with functional vasculature, avoiding the current problem of necrotic core and cell death inside the organoid. Culturing organoids for longer periods of time also presents other advantages such as the development of more cell types with better maturity, therefore creating a better complex 3D model. For instance, oligodendrocytes only start appearing at around 4 months, reaching maturity after 5–6 months [226].

However, it remains to be determined how the complex environment in utero, with exposure to the mother’s pregnancy hormones and other factors, could one day be included in such a model, in hopes of understanding whether such early life events can have a major impact on later neurodegenerative diseases. Furthermore, many unknowns remain regarding the action of sex hormones in males compared to females. Since both sexes share the same sex hormones, though in different physiological levels, makes it difficult to understand their differing mechanisms of action. We first need to gain more knowledge on this topic if we hope to one day be able to mimic it in a dish. Lastly, while sex hormones are highly relevant during development and provide neuroprotective features, the epigenetic drift during aging and reactivation of microglia-specific genes could be even more relevant for the sex specific differences seen in AD.

## Figures and Tables

**Figure 1 biomedicines-11-01261-f001:**
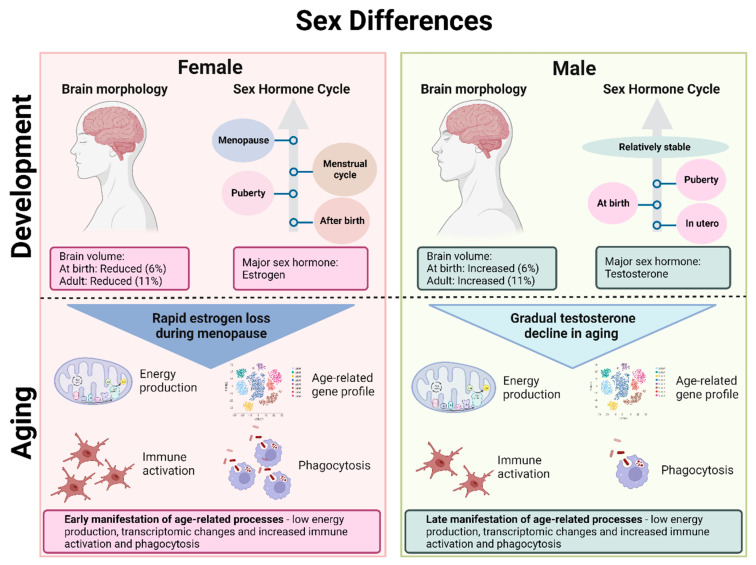
Sex differences in the brain during development and aging. The female brain has a reduced volume compared to the male brain, and the major sex hormone in the female brain is estrogen compared to testosterone in the male brain. In females, there is an increase in estrogen after birth, another increase at puberty, varying levels of sex hormones during the menstrual cycle and a drastic drop in estrogen at menopause. In males, testosterone is increased in utero, at birth and during puberty, staying relatively stable throughout adulthood. During aging, females therefore experience a rapid estrogen loss compared to the gradual decline in testosterone in men. Moreover, females experience an earlier manifestation of age-related processes such as low energy production, transcriptomic changes and immune responses compared to males.

**Figure 2 biomedicines-11-01261-f002:**
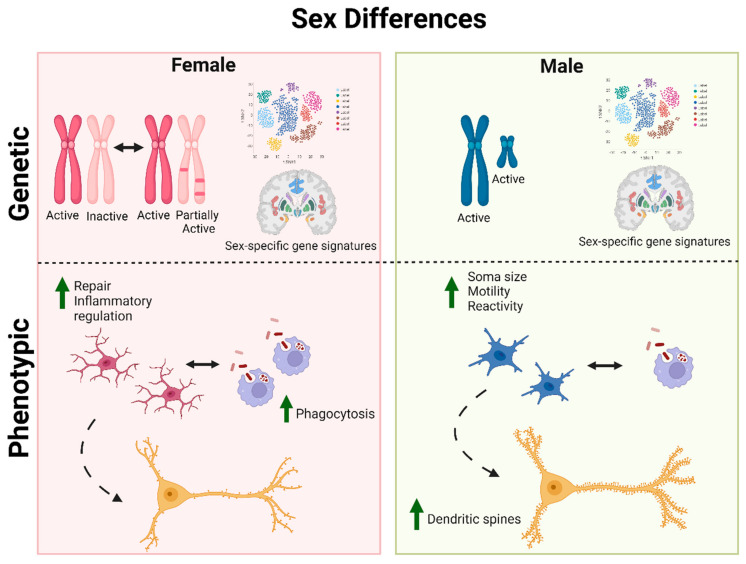
Sex differences in microglia. X inactivation should lead to an equal expression of X chromosome genes in both sexes. However, some genes escape this inactivation, which contributes to sex-specific gene signatures and intrinsic differences in female compared to male microglial immune responses. Sex differences in microglial transcriptomes are observed throughout development. Phenotypically, male microglia are generally more reactive with larger somas and increased motility, whereas female microglia are more phagocytic with a profile consistent with repair and inflammatory regulation. Male microglia further contribute to an increased density of dendritic spines and male-specific behavior.

**Figure 3 biomedicines-11-01261-f003:**
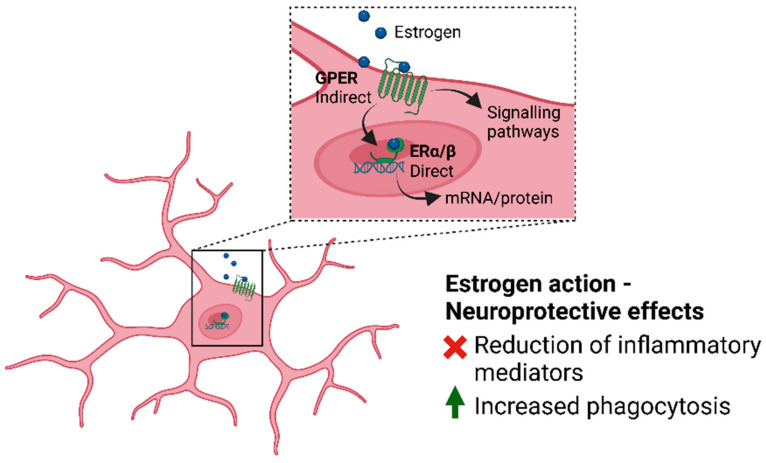
Estrogen action and receptors. The main estrogen receptors are the nuclear estrogen receptors (ERα and ERβ). Upon estrogen binding, these can act as a transcription factor, directly influencing gene expression. Additionally, estrogen can act on membrane-bound receptors, such as G-protein coupled estrogen receptors (GPER), through non-genomic mechanisms via protein kinase cascades. Estrogen binding to either of these promotes phagocytosis and an anti-inflammatory microglia phenotype, mediated by *ANXA1,* ultimately possessing neuroprotective effects.

**Figure 4 biomedicines-11-01261-f004:**
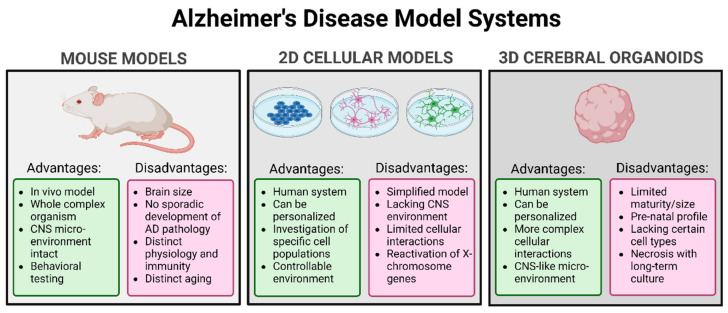
Advantages and disadvantages of commonly used AD model systems.

**Figure 5 biomedicines-11-01261-f005:**
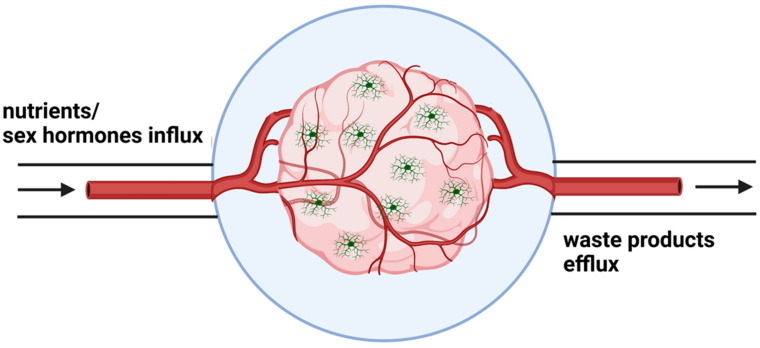
Advanced microfluidic system with a supporting vascularization to cerebral organoids would allow proper oxygen and nutrient flow to the whole organoid as well as removal of waste products, greatly facilitating long-term cultures. Additionally, incorporating microglia to have an immuno-competent cerebral organoid would lead to a complex model system. Investigating the role of sex hormones in development and disease would then be possible as female-derived organoids could be cultured until a post-natal stage when estrogen could be administrated to mimic in vivo exposure. This could further be compared to male-derived cerebral organoids exposed to testosterone in a manner also mimicking in vivo exposure.

**Table 1 biomedicines-11-01261-t001:** Single nucleotide polymorphisms (SNPs) associated with Alzheimer’s disease; SNP ID, genetic location, allelic variant, populational frequency, function and main cellular expression in the brain.

SNP ID	Gene	Variant	Frequency	Function	Brain Expression
rs3752246	ABCA7	G > C/C > T	G = 0.15755 (C = 0.84245)	Lipid homeostasis	Neurons, astrocytes, microglia and oligodendrocytes
rs3764650	ABCA7	T > G	G = 0.102719 (T = 0.897281)	Lipid homeostasis	Neurons, astrocytes, microglia and oligodendrocytes
rs1800764	ACE	C > G/C > T	C = 0.481027 (T = 0.518973)	Blood pressure and electrolyte balance	Neurons, microglia and oligodendrocytes
rs442495	ADAM10	T > C/T > G	C = 0.40762 (T = 0.59238)	Lipid transport and phagocytosis	Oligodendrocytes, microglia, neurons and astrocytes
rs7412	APOC1	C > T	T = 0.083116 (C = 0.916884)	Lipid transport	Microglia, astrocytes and neurons
rs429358	APOC1	T > C	C = 0.074418 (T = 0.925582)	Lipid transport	Microglia, astrocytes and neurons
rs4663105	BIN1	A > C	C = 0.44298 (A = 0.55702)	Endocytosis	Oligodendrocytes, microglia, neurons and astrocytes
rs744373	BIN1	A > C/A > G/A > T	G = 0.297771 (A = 0.702229)	Endocytosis	Oligodendrocytes, microglia, neurons and astrocytes
rs9381563	CD2AP	C > A/C > G/C > T	C = 0.39767 (T = 0.60233)	Cytoskeleton regulation	Microglia, neurons, astrocytes and oligodendrocytes
rs9349407	CD2AP	G > C	C = 0.24780 (G = 0.75220)	Cytoskeleton regulation	Microglia, neurons, astrocytes and oligodendrocytes
rs10948363	CD2AP	A > G	G = 0.24596 (A = 0.75404)	Cytoskeleton regulation	Microglia, neurons, astrocytes and oligodendrocytes
rs1354106	CD33	T > G	G = 0.34719 (T = 0.65281)	Cell adhesion and immune cell homeostasis	Microglia
rs3865444	CD33	C > A	A = 0.302324 (C = 0.697676)	Cell adhesion and immune cell homeostasis	Microglia
rs12459419	CD33	C > A/C > G/C > T	T = 0.26380 (C = 0.73620)	Cell adhesion and immune cell homeostasis	Microglia
rs4845378	CHRNB2	G > A/G > T	T = 0.08354 (G = 0.91646)	Ion transport	Neurons, oligodendrocytes and astrocytes, microglia
rs11136000	CLU	T > A/T > C	C = 0.607739 (T = 0.392261)	Apoptosis, complement and immunity	Astrocytes, microglia, neurons and oligodendrocytes
rs9331896	CLU	C > A/C > G/C > T	C = 0.42127 (T = 0.57873)	Apoptosis, complement and immunity	Astrocytes, microglia, neurons and oligodendrocytes
rs1532278	CLU	T > A/T > C	T = 0.374146 (C = 0.625854)	Apoptosis, complement and immunity	Astrocytes, microglia, neurons and oligodendrocytes
rs4236673	CLU	A > C/A > G	A = 0.33623 (G = 0.66377)	Apoptosis, complement and immunity	Astrocytes, microglia, neurons and oligodendrocytes
rs3818361	CR1	A > C/A > G	A = 0.201980 (G = 0.798020)	Complement and immunity	Oligodendrocytes and microglia
rs6656401	CR1	A > G/A > T	A = 0.179434 (G = 0.820566)	Complement and immunity	Oligodendrocytes and microglia
rs4844609	CR1	A > G/A > T	A = 0.021785 (T = 0.978215)	Complement and immunity	Oligodendrocytes and microglia
rs1064039	CST3	C > G/C > T	T = 0.19579 (C = 0.80421)	Protease inhibitor	Astrocytes, oligodendrocytes and microglia
rs11771145	EPHA1	G > A/G > T	A = 0.351605 (G = 0.648395)	Angiogenesis and cell adhesion	Neurons and microglia
rs11763230	EPHA1	C > T	T = 0.20866 (C = 0.79134)	Angiogenesis and cell adhesion	Neurons and microglia
rs11767557	EPHA1	T > A/T > C	C = 0.194037 (T = 0.805963)	Angiogenesis and cell adhesion	Neurons and microglia
rs10793294	GAB2	C > A/C > G	C = 0.251454 (A = 0.748546)	Cell proliferation and growth	Oligodendrocytes, microglia, neurons and astrocytes
rs3745833	GALP	C > A/C > G/C > T	G = 0.33124 (C = 0.66875)	Neuropeptide	Neurons
rs6931277	HLA-DQA1	A > T	T = 0.14976 (A = 0.85024)	Adaptive immunity	Microglia
rs1143634	IL1B	G > A	A = 0.227942 (G = 0.772058)	Inflammatory response	Microglia
rs35349669	INPP5D	C > T	T = 0.38287 (C = 0.61713)	Apoptosis, lipid metabolism and immunity	Microglia
rs10933431	INPPD5	G > C	G = 0.30651 (C = 0.69349)	Apoptosis, lipid metabolism and immunity	Microglia
rs190982	MEF2C	G > A/G > C	G = 0.376709 (A = 0.623291)	Apoptosis, differentiation, neurogenesis and immune proliferation	Neurons, microglia, oligodendrocytes and astrocytes
rs190982	MEF2C	G > A/G > C	G = 0.376709 (A = 0.623291)	Apoptosis, differentiation, neurogenesis and immune proliferation	Neurons, microglia, oligodendrocytes and astrocytes
rs558678	MS4A2	T > G	G = 0.24086 (T = 0.75914)	Immune responses	Neurons
rs4938933	MS4A4A	C > G/C > T	C = 0.404463 (T = 0.595537)	Signal transduction and immune responses	Microglia
rs610932	MS4A4A	T > C/T > G	T = 0.430441 (G = 0.569559)	Signal transduction and immune responses	Microglia
rs10897011	MS4A4E	G > A	A = 0.33764 (G = 0.66236)	Innate immunity	Microglia
rs610932	MS4A6A	T > C/T > G	T = 0.430441 (G = 0.569559)	Signal transduction and immune responses	Microglia
rs7935829	MS4A6A	A > G	G = 0.35879 (A = 0.64121)	Signal transduction and immune responses	Microglia
rs2081545	MS4A6A	C > A/C > T	A = 0.21413 (C = 0.78587)	Signal transduction and immune responses	Microglia
rs11754661	MTHFD1L	G > A/G > T	A = 0.061583 (G = 0.938417)	Metabolism	Microglia, neurons, oligodendrocytes and astrocytes
rs3800324	PGBD1	G > A	A = 0.054414 (G = 0.945586)	Unknown	Neurons, oligodendrocytes, astrocytes and microglia
rs3851179	PICALM	T > C	T = 0.361052 (C = 0.638948)	Endocytosis	Oligodendrocytes, microglia, neurons and astrocytes
rs561655	PICALM	G > A/G > T	G = 0.348713 (A = 0.651287)	Endocytosis	Oligodendrocytes, microglia, neurons and astrocytes
rs541458	PICALM	C > T	C = 0.315402 (T = 0.684598)	Endocytosis	Oligodendrocytes, microglia, neurons and astrocytes
rs10792832	PICALM	A > C/A > G/A > T	A = 0.29879 (G = 0.70121)	Endocytosis	Oligodendrocytes, microglia, neurons and astrocytes
rs2058716	PRKD3	G > A/G > C	G = 0.31978 (C = 0.68022)	Apoptosis and lipid transport	Microglia, oligodendrocytes, neurons and astrocytes
rs165932	PSEN1	G > A/G > T	G = 0.424510 (T = 0.575490)	Amyloidogenic- and Notch processing	Oligodendrocytes, neurons, microglia and astrocytes
rs28834970	PTK2B	T > C	C = 0.31596 (T = 0.68404)	Adaptive immunity	Neurons, microglia, oligodendrocytes and astrocytes
rs2301275	PVR	A > C/A > G	G = 0.23252 (A = 0.76748)	Cell adhesion	Neurons, oligodendrocytes, astrocytes and microglia
rs142787485	RAB10	A > G	G = 0.02890 (A = 0.97110)	Protein transport	Oligodendrocytes, neurons, microglia and astrocytes
rs2376866	RELB	C > T	T = 0.14727 (C = 0.85273)	Apoptosis, cell growth and immunity	Microglia, neurons, oligodendrocytes and astrocytes
rs11218343	SORL1	T > A/T > C	C = 0.02728 (T = 0.97272)	Endocytosis	Microglia, neurons, astrocytes and oligodendrocytes
rs2282649	SORL1	C > A/C > T	T = 0.28928 (C = 0.71072)	Endocytosis	Microglia, neurons, astrocytes and oligodendrocytes
rs2306604	TFAM	A > C/A > G/A > T	G = 0.41883 (A = 0.58117)	Transcription regulation	Oligodendrocytes, neurons, astrocytes and microglia
rs5011436	TMEM106B	A > C/A > G/A > T	C = 0.27232 (A = 0.72768)	Lysosomal transport	Neurons, oligodendrocytes, astrocytes and microglia
rs187370608	TREM2	G > A	A = 0.00201 (G = 0.99799)	Phagocytosis and immune activation	Microglia
rs75932628	TREM2	C > A/C > T	T = 0.00232 (C = 0.99768)	Phagocytosis and immune activation	Microglia

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
