# Peer review of "Complexity of Sex Differences and Their Impact on Alzheimer’s Disease"

_biomedicines, 2023, doi:10.3390/biomedicines11051261_

Round 1

Reviewer 1 Report

This is an interesting and comprehensive review article on the impact of sex differences on Alzheimer disease. Although this issue is also discussed in other recent review articles (e.g doi: 10.3390/ijms24043205, doi: 10.1007/7854_2022_408, DOI: 10.3389/fnins.2022.954999, doi: 10.1016/j.jsbmb.2012.11.006), in the present review article, authors summarizes many aspects of neuronal physiology affecting by sex hormones, including brain development, aging, microglia function, immune responses, Alzheimer disease onset and progression. Moreover, available models for studying Alzheimer disease are discussed in relation to their applicability in studying sex differences and microglia involvement in the disease. This facet is of particular interest and is only slightly or not at all covered in other reviews. The article is well written and could be published as is.

 Minor comment

The above-mentioned citations could be cited in this review article

Author Response

Reviewer 1

Comments and Suggestions for Authors

This is an interesting and comprehensive review article on the impact of sex differences on Alzheimer disease. Although this issue is also discussed in other recent review articles (e.g doi: 10.3390/ijms24043205, doi: 10.1007/7854_2022_408, DOI: 10.3389/fnins.2022.954999, doi: 10.1016/j.jsbmb.2012.11.006), in the present review article, authors summarizes many aspects of neuronal physiology affecting by sex hormones, including brain development, aging, microglia function, immune responses, Alzheimer disease onset and progression. Moreover, available models for studying Alzheimer disease are discussed in relation to their applicability in studying sex differences and microglia involvement in the disease. This facet is of particular interest and is only slightly or not at all covered in other reviews. The article is well written and could be published as is.

Minor comment

The above-mentioned citations could be cited in this review article 

We thank the reviewer for the comment and suggestion of additional citations. Two of the suggested papers have now been cited in the manuscript (line 28 and 343) and added to the reference list (line 826 and 973) as suggested by the reviewer (doi: 10.3390/ijms24043205, doi: 10.1007/7854_2022_408).

Reviewer 2 Report

In this review, authors summarize the differences that exist between both sexes in the brain morphology, the influence of sex hormones in the brain, and how aging affects the brain in a sex-dependent manner. They also highlight microglia as major player in the neuroinflammation in Alzheimer’s disease (AD).

Overall, the review seems to be out of focus. A reader may expect to find a summary of the mechanisms that drive differences in AD progression between both sexes. However, authors aimed to cover very complex aspects influenced by sex, such as brain development and morphology, which each one would need a separate review.

Specifically, the section 3.1 could be reduced from 1.5 pages to a few sentences as according to the text the brain morphology is not relevant for sex differences in AD.  

In the section 3.2, is there evidence that sex hormones during early stages of life influence sex differences in AD?

The section 3.3 is mostly focused on hormone decline. Could be other mechanisms involved in sex differences during aging beyond sex hormones? What about escapee genes from the X chromosome inactivation?

Authors should make clear why they want to spend five pages talking about microglia as major effectors of sex differences in AD. Neuroinflammation is not only caused by microglia. Circulating immune cells, astrocytes, macrophages, etc. also participate in neuroinflammation in AD.

Line 332: “women AD patients are usually diagnosed at an advanced age and therefore aging mechanisms, previously discussed…”. Indeed, the review does not cover the mechanisms that may drive higher prevalence of AD in women than in men. Sex differences in metabolism, autophagy, gene transcription, genome stability, etc. contribute to sex-biased AD progression.

In the section 5.2, authors discuss different methods to study AD. However, they do not mention what methods are available to study the effect of sex in AD. Some methods could be the Four Core Genotype mice, Y* mice, hormone capsules…

Author Response

Reviewer 2

In this review, authors summarize the differences that exist between both sexes in the brain morphology, the influence of sex hormones in the brain, and how aging affects the brain in a sex-dependent manner. They also highlight microglia as major player in the neuroinflammation in Alzheimer’s disease (AD).

Overall, the review seems to be out of focus. A reader may expect to find a summary of the mechanisms that drive differences in AD progression between both sexes. However, authors aimed to cover very complex aspects influenced by sex, such as brain development and morphology, which each one would need a separate review. 

In this review we aimed to provide a systematic overview of sex differences that might underlie the predisposition of AD in women. Therefore, we decided to include a number of differences between males and females, to point out the complexity of sex differences, in which multiple aspects should be considered when studying disease mechanisms in AD. We agree with the reviewer that some of the topics could be elaborated in separated reviews, but the scope of the current manuscript was to create an overall summary of sex differences that could contribute to the high prevalence of AD in women compared to men.

Specifically, the section 3.1 could be reduced from 1.5 pages to a few sentences as according to the text the brain morphology is not relevant for sex differences in AD.  

Although we agree with the reviewer that sex differences in brain morphology is not the only contributing factor to the differences of AD prevalence in women compared to men, we still believe that the differences in brain volume and connectivity is relevant for sex differences in AD. This is a systemic review describing a wide range of potential underlying factors, with the aim of discussing multiple aspects relevant for the sex bias in AD. For example, it is clear by the literature that there is a higher connectivity in the DMN in females and that this region is directly impaired in AD. These studies indicate a relevant role of brain morphology sex differences in the disease. Moreover, we are clearly stating that changes in brain morphology are not the solitary contributor to the sex bias, which are most likely caused by a complex interplay between sex hormone levels, genetic expression signature, and X chromosome gene dosage, which should all be considered. We have therefore decided to leave the section as it is.

In the section 3.2, is there evidence that sex hormones during early stages of life influence sex differences in AD? 

To our knowledge there is no evidence that sex hormones during early stages of life influence sex differences in AD. However, the aim of this section was to provide insight into the differences in sex hormone cycle and the distinct difference in major type of sex hormone between males and females which thus becomes relevant also in the development in AD.

The section 3.3 is mostly focused on hormone decline. Could be other mechanisms involved in sex differences during aging beyond sex hormones? What about escapee genes from the X chromosome inactivation?

We thank the reviewer for raising this question. Section 3.3 is mostly discussing the role of sex hormones during aging and its effect on AD progression. We acknowledge that other mechanisms could be involved in sex differences during aging, but we have decided to focus this part on sex hormones, due to the clear differences in hormone level decline that is observed during aging. However, the potential involvement of escapee genes from X chromosome inactivation is mentioned in section 4.1 Male and female microglia and has now been further elaborated (line 395-404). We have decided to include the potential contribution of escapees in this section as many of the genes prone to escape X chromosome inactivation are involved in immune responses and are thus highly expressed by microglia, providing a relevant link between sex differences, microglia and neuroinflammation. 

Authors should make clear why they want to spend five pages talking about microglia as major effectors of sex differences in AD. Neuroinflammation is not only caused by microglia. Circulating immune cells, astrocytes, macrophages, etc. also participate in neuroinflammation in AD.

We appreciate the reviewer’s comments that other cell types such as astrocytes are important contributors of neuroinflammation. However, with microglia being at the center of current investigations as inflammatory drivers of the disease, highlighted by the mentioned identification of risk SNPs in genes enriched in microglia, we have decided to focus on this cell population. Moreover, sex differences are particularly relevant in microglia as many immune related genes are located on the X chromosome. In light of this, recent efforts are placed on studying the microglial contribution to AD (line 93-94). We have now also made it clear already in the abstract that the focus of this review will be on microglia (line 16).

Line 332: “women AD patients are usually diagnosed at an advanced age and therefore aging mechanisms, previously discussed…”. Indeed, the review does not cover the mechanisms that may drive higher prevalence of AD in women than in men. Sex differences in metabolism, autophagy, gene transcription, genome stability, etc. contribute to sex-biased AD progression.

We agree that multiple mechanisms might underlie the higher prevalence of AD in women. In this review we have provided a systemic summary of some of these mechanisms. Unfortunately, we are unable to cover all potential sex-related disease mechanisms in one review. 

In the section 5.2, authors discuss different methods to study AD. However, they do not mention what methods are available to study the effect of sex in AD. Some methods could be the Four Core Genotype mice, Y* mice, hormone capsules…

In section 5.2 we are presenting different systems that are commonly used to model AD. The applicability of these model systems in studying the effect of sex in AD is elaborated in the discussion section, and advantages and disadvantages with each model are described. As suggested by the reviewer we have now included the potential use of Four core genotype and XY* models (line 682-687).

Reviewer 3 Report

This an excellent and timely review of the literature review of the literature.

There are two apparent gaps:

1) in discussing animal models of AD the authors do not tackle sex differences. The in vitro and organoid models also do not address how you would study sex differences in this context. I would urge the authors to address these explicitly in the manuscript. 

2) While the focus is predominately on microglial sex differences, the authors should also address other potential sex effects of other cell types, for example endothelial cells, oligiodendrocytes etc. Several paragraphs acknowledging these and other cells and potential sex differences would be a great addition to an already outstanding review.

Author Response

Reviewer 3

This an excellent and timely review of the literature review of the literature. 

There are two apparent gaps:

1) in discussing animal models of AD the authors do not tackle sex differences. The in vitro and organoid models also do not address how you would study sex differences in this context. I would urge the authors to address these explicitly in the manuscript. 

We thank the reviewer for addressing this point. We have now included a small section introducing the four core genotype and XY* mice models that can be used to investigate sex differences (line 682-687). For both in vitro and organoid model systems, we have highlighted that cell lines originating from both male and female patients/controls should be included in the experimental set-up to account for potential sex differences (line 719-721). Furthermore, we propose that sex hormones could be added to organoids in order to recapitulate in vivo exposure (line 774-777).

2) While the focus is predominately on microglial sex differences, the authors should also address other potential sex effects of other cell types, for example endothelial cells, oligiodendrocytes etc. Several paragraphs acknowledging these and other cells and potential sex differences would be a great addition to an already outstanding review.

Although the reviewer is making an important point, we have unfortunately not included other cell types. Due to the complexity and length of the review we found it appropriate to put the focus towards a single cell type. Microglia are at the center of current investigations as proinflammatory drivers of AD, and many risk SNPs are located in genes highly enriched in microglia. Moreover, many of these immune-related genes are located on the X chromosome. Therefore, we have decided to only focus on microglia in the present review. 

Round 2

Reviewer 2 Report

What is the contribution of estrogen depletion on AD?

Could authors emphasize the relevance of this review on the AD field?

What authors would conclude as major drivers of sex differences in AD?

Author Response

What is the contribution of estrogen depletion on AD?

Response 1: We thank the reviewers to point out that this section needs additional information. We have now added a section discussing the effects of hormone replacement therapy and tamoxifen induced ablation of estrogen (line 251-265).

Could authors emphasize the relevance of this review on the AD field?

Response 2: Thank you for pointing this out. We have elaborated on the need to understand sex differences in AD microglia in the light of new pharmacological intervention studies implementing immunomodulating drugs (line 669-676).

What authors would conclude as major drivers of sex differences in AD?

Response 3: We appreciate the reviewer's comments. We have inserted concluding sentences in the end of the review. We conclude that the epigenetic drift and reactivation of silenced genes on the X chromosome are important biological events, which might be the culprit for the increased incidence of AD in women (line 829-832).

Round 3
